# Tephritid Fruit Fly Species Composition, Seasonality, and Fruit Infestations in Two Central African Agro-Ecological Zones

**DOI:** 10.3390/insects13111045

**Published:** 2022-11-13

**Authors:** Samuel Nanga Nanga, Rachid Hanna, Apollin Fotso Kuate, Komi K. M. Fiaboe, Ibrahim Nchoutnji, Michel Ndjab, Désiré Gnanvossou, Samira A. Mohamed, Sunday Ekesi, Champlain Djieto-Lordon

**Affiliations:** 1International Institute of Tropical Agriculture (IITA), Messa, Yaoundé P.O. Box 2008, Cameroon; 2Center for Tropical Research, Institute of the Environment and Sustainability, University of California, Los Angeles, CA 90095, USA; 3Institute of Agricultural Research for Development, Foumbot P.O. Box 163, Cameroon; 4International Institute of Tropical Agriculture (IITA), Tri Postal 08 P.O. 0932, Benin; 5International Centre of Insect Physiology and Ecology (*icipe*), Nairobi P.O. Box 30772-00100, Kenya; 6Department of Animal Biology and Physiology, Faculty of Science, University of Yaounde I, Yaoundé P.O. Box 812, Cameroon

**Keywords:** *Bactrocera dorsalis*, *Ceratitis cosyra*, *Ceratitis anonae*, male lure, food-bait, host plant records

## Abstract

**Simple Summary:**

Tephritid fruit flies are a major threat to fruit production in sub-Saharan Africa. Central Africa has lagged considerably behind the rest of the world in fruit fly knowledge and management. In six consecutive years of research, we developed new knowledge on the diversity, seasonality, attraction to various lures/baits, and fruit infestations of frugivorous fruit fly infesting fruits, particularly mango and guava, in two contrasting agro-ecological zones (AEZs) of Cameroon representing the highland and mid-altitude AEZs of Central Africa’s Congo Basin. Ten fruit fly species from four genera—*Bactrocera*, *Ceratitis*, *Dacus*, and *Perilampsis*—were found in traps and fruits in both AEZs. Overall, the exotic fruit fly *Bactrocera dorsalis* was most abundant in traps and in fruits, particularly mango and guava, followed by the native *Ceratitis cosyra* and *C. anonae*, which were the dominant *Ceratitis* species in mid-altitude and highland AEZs, respectively. As expected, seasonal patterns of the three species largely followed rainfall and fruit availability. Of the three food baits used, Torula yeast was the most efficient in trapping all species, compared with BioLure and Mazoferm. Among the 25 sampled fruit species, *Irvingia wombolu*, *Dacryodes edulis*, *Voacanga africana* and *Trichoscypha abut* were new worldwide host records for *B. dorsalis*.

**Abstract:**

*Bactrocera dorsalis* and several Africa-native *Ceratitis* species are serious constraints to fruit production in sub-Saharan Africa. A long-term trapping and fruit collection study was conducted (2011–2016) in two contrasting agro-ecological zones (AEZs) of Cameroon to determine fruit fly species composition, seasonality, attraction to various lures and baits, and fruit infestation levels. Ten tephritid species from genera *Bactrocera*, *Ceratitis*, *Dacus*, and *Perilampsis* were captured in traps. *Bactrocera dorsalis* was the most dominant of the trapped species and persisted throughout the year, with peak populations in May–June. *Ceratitis* spp. were less abundant than *B. dorsalis*, with *Ceratitis anonae* dominating in the western highland zone and *Ceratitis cosyra* in the humid forest zone. Methyl eugenol and terpinyl acetate captured more *B. dorsalis* and *Ceratitis* spp., respectively than Torula yeast. The latter was the most effective food bait on all tephritid species compared with BioLure and Mazoferm. *Bactrocera dorsalis* was the dominant species emerging from incubated fruits, particularly mango, guava, and wild mango. Four plant species—*I. wombolu*, *Dacryodes edulis*, *Voacanga Africana* and *Trichoscypha abut*—were new host records for *B. dorsalis*. This study is the first long-duration and comprehensive assessment of frugivorous tephritid species composition, fruit infestations, and seasonality in Central Africa.

## 1. Introduction

Worldwide, tephritid fruit flies (Diptera: Tephritidae) are among the world’s most economically important crop pests, with at least 200 pest species [1,2]. In sub-Saharan Africa, several highly polyphagous Africa-native species—belonging to the genera *Ceratitis* Macleay and *Dacus* Fabricius have been recognized as economically important pests of several cultivated and wild fruit species, particularly mango (*Mangifera indica* L. (Sapindales: Anacardiaceae), guava (*Psidium guajava* (L.) (Myrtales: Myrtaceae)), citrus (Citrus spp. L. (Sapindales: Rubiaceae)), and several cucurbit and solanaceous vegetables [3,4,5].

The traditional problems with tephritid fruit flies have been aggravated in recent years by the invasion of the African continent by the Oriental fruit fly *Bactrocera dorsalis* (Hendel) (Diptera: Tephritidae), which was first detected in coastal Kenya in 2003 [6] and has spread to at least 32 countries in continental Africa and adjacent island countries [7]. Since the detection of *B. dorsalis* in Africa, several studies have established the African host range of this species and quantified crop losses (exceeding 57%) due to its infestations [8,9,10]. At present, *B. dorsalis* has been found infesting fruits of 40 host plant species, with mango, guava, citrus, and loquat (*Eriobotrya japonica* (Thunb.) Lindl. (Rosales: Rosaceae)) being the major infested cultivated hosts [11,12,13,14]. In addition to causing extensive fruit losses in the field, fruit flies greatly restrict mango and other host fruit exports from Africa, particularly to the European Union (EU), which, for example, intercepted and rejected more than 141 shipments of Cameroonian mango from 2011 to 2018 [15], resulting in substantial financial losses. While several control methods have been developed and deployed across the continent [8,16,17,18,19,20], the large majority of fresh fruit producers in Cameroon and throughout Central Africa continue to experience substantial yield losses caused by fruit flies and they do not yet have the necessary resources and knowledge to successfully use available and new fruit fly control methods. The prevailing agronomic and plant protection practices are of very low or no input type. Apart from the common occasional weeding, pesticide and fertilizer inputs are rare.

Knowledge of fruit fly species composition and their respective seasonal abundance using complementary monitoring tools in relation to host plant phenology under different environments is crucial to the understanding of population dynamics of these insects and the subsequent development and implementation of interventions to limit their infestations and damage [10,21]. Such knowledge is predicated on proper fruit fly species identification and quantification of the levels of host infestation which are fundamental for establishing the economic status of the pests and ultimately for developing and adopting effective pest control interventions [22,23] Two approaches have been traditionally used to provide the aforementioned needed information: (1) effective tools based on food baits and male lures for monitoring and estimating the abundance of adult fruit flies, and (2) systematic fruit samplings to determine host range and quantify the levels and rates of fruit infestations by the various fruit fly species present in the systems. The latter is often complemented with random fruit sampling from areas outside the targeted cultivated fields to determine the fruit fly host range [23,24]. Ideally, monitoring tools and host fruit infestations should be tested and used over several years and in multiple environments to establish sufficient details of the bio-ecological context where management options will be developed and implemented.

Several commercially available male lures and food baits have been developed and used widely for fruit fly monitoring, but their performance has been shown to vary with factors such as climate, fruit fly species, and other factors that affect fruit fly populations [9,25,26,27,28,29]. All available studies in Africa are from several agro-ecologies, but none are from the mid-altitude, high rainfall agro-ecologies that are prevalent in much of the Congo Basin of Central Africa.

The male lures methyl eugenol and terpinyl acetate are known to, respectively, attract *Bactrocera* and *Ceratitis* species, while Culure is known to attract various (though not all) *Dacus* species [24,30,31]. For principally females, several food baits including Torula yeast, Mazoferm, and Nulure have been developed and used to attract and monitor several fruit fly species [25,28,30,31,32]. To our knowledge, monitoring the performance of food baits and male lures on fruit flies under the various environments that are prevalent in Central Africa, is lacking. Similarly, compared with other regions of Africa, information on fruit fly species composition, host range, crop losses, and seasonality, as well as various trapping approaches and control measures in Central Africa, are scarce.

The Congo Basin of Central Africa harbours a rich humid forest with a high diversity of wild fruit trees that could, at the same time, represent a reservoir for fruit flies and their natural enemies. Central Africa further includes the five key agro-ecologies encountered across the African continent, from desert and arid agro-ecologies to dense high-rainfall and humid forest zones [33]. Preliminary information from fruit collections in Cameroon revealed the presence of several species including *B. dorsalis*, *Ceratitis cosyra* (Walker), *Ceratitis anonae* (Graham), *Ceratitis capitata* (Wiedemann), *Ceratitis quinaria* (Bezzi), *Dacus punctatifrons* Karschand *Dacus bivittatus* (Bigot) [13,34,35,36,37] Continuing to be scarce, however, is multi-year quantitative information on fruit fly’s species composition and their seasonal dynamics, host utilization, and fruit infestation levels, particularly from the principal commercial fruit species mango and guava, and the performance of different monitoring tools in mid-altitude humid and high rainfall agro-ecologies from Central Africa.

The broad objective of this study is to establish and validate basic multi-year data necessary for the development of integrated pest management programs of fruit flies across two agro-ecological zones in Cameroon with contrasting climate and farming systems, representing a cross-section of the mid-altitude agro-ecologies of Central Africa. The study has the following specific objectives: (1) determine the diversity of fruit fly species and the level of their infestation of mango, guava and other common fruit hosts; (2) compare the performance of male lures and food baits for monitoring the abundance and seasonality of fruit flies in mango and mixed fruit orchards; and (3) determine the contribution of temperature, relative humidity, and rainfall amount to the variation in fruit fly abundance. The results from this Cameroon study can possibly be extended to the rest of the Congo Basin, as the southern half of Cameroon is widely considered agro-ecologically a close representative of much of the rest of Central Africa.

## 2. Materials and Methods

### 2.1. Study Sites

The study was conducted over 5–6 years (from January 2011 to December 2016) in 2 agro-ecological zones (AEZs) of Cameroon as delimited by the Cameroon Institute of Agricultural Research for Development (IRAD) [38] (Figure 1). The two target AEZs included the western highlands, with a mono-modal rainfall pattern (WH-MR), and humid forest, with bimodal rainfall (HF-BR) (Figure 1). Both AEZs differed in their topography, climate characteristics, and cropping systems [39]. The choice of the two zones was based on the richness, diversity and availability of fruit tree species. One experimental site was established in each AEZ for fruit fly trapping using food baits and male lures, and for evaluation of fruit infestations by fruit flies. Because of the long-term nature of the experiments and the need to secure traps for continuous monitoring over a period of 6 years, the traps were installed in the experimental orchard of the IRAD research station in Foumbot, for the WH-MR, and at the International Institute of Tropical Agriculture (IITA) in Nkolbisson, for the HF-BR (Figure 1). Each orchard was characterized according to the description of the area, climatic conditions, fruit species present and management options (Table 1). A homogenous hectare of mango was used in Foumbot, while a mixed hectare of fruit species was selected in Nkolbisson (Table 1).

In addition to the two experimental sites, fruits were collected from 6 other locations in the WH-MF within a 70 km radius of Foumbot, and 5 other locations in HF-BR, within a 70 km radius of Nkolbisson (Figure 1). The selection of the fruit collection sites was based on the presence of mango or guava orchards at or in the vicinity of the collection sites.

### 2.2. Male Lures and Food Baits Traps

Male lures and food baits used for fruit fly trapping are described in Table 2. In the studied orchards, Multilure traps were used for food baits and bucket traps were used for male lures. The Multilure traps (Better World Manufacturing, Inc., Fresno, CA, USA) were made of a yellow-colored plastic base and a transparent plastic upper part [28], while bucket traps were similar to a Tephritrap^®^ (Pherobank, Wageningen, The Netherlands), and were made of a yellow-colored cylindrical plastic container with four equidistant holes at the upper third, and a white-colored lid.

All traps were suspended from tree branches with a galvanized steel wire at ~2 m above ground and at least 20 m apart. The wire was coated at its middle length with a thick layer of Tanglefoot (The Tanglefoot Co., Grand Rapids, MI, USA) to prevent cursorial access to the traps by predators, particularly the common weaver ant *Oecophylla longinoda* (Latreille). The number of traps varied according to the number of attractants used. For this purpose, 2 and 4 traps of each attractant were installed, respectively, at the Nkolbisson and Foumbot sites, for a total of 10 traps in Nkolbisson and 12 traps in Foumbot.

For male lure-based traps, a dental cotton roll soaked with 2 mL of either methyl eugenol or terpinyl acetate was suspended from the centre of the trap lid. Two, 5 cm strips impregnated with 2, 2-Dimethyl dichlorovinyl phosphate (DDVP) (Hercon Environmental Corporation, Engsville, PA) were placed at the bottom of the trap as the killing agent. For food-bait traps, BioLure’s 3 components, packed individually in a sachet containing either ammonium acetate, trimethylamine, or putrescine, were adhered to the inside of the Multilure trap. Two DDVP strips were placed at the bottom of each trap as a killing agent. Torula yeast was used as a liquid bait consisting of 2 pellets (~8 g total containing 3% borax) dissolved in 350 mL of water per trap [28]. Similarly, the commercial product Mazoferm was diluted in 350 mL of water to obtain a 6% concentration, with 2 g of borax added to the solution as a preservative [28].

Food bait and male lure traps were inspected in both orchards at weekly intervals. Torula yeast and Mazoferm baits were replaced weekly, while BioLure, male lures, DDVP strips, and cotton rolls were renewed monthly [26,28,40]. Trap servicing techniques and regular rotation among trees followed those of [28]. All the specimens were transferred and preserved in vials containing 70% ethanol. All samples were brought to the Entomology Laboratory of IITA in Yaoundé for identification.

Meteorological data were collected at each location with a Hobo Pro v2 data logger for temperature and RH (Onset Computing, Bourne, MA, USA), and a Tru-Chek^®^ Direct-Reading rain gauge (Forestry Suppliers, Jackson, MS, USA). Temperature and RH were recorded at hourly intervals and the data were retrieved at monthly intervals, while rain amounts were collected between 7 and 8 am daily throughout the study periods.

### 2.3. Host Fruit Collection and Handling

Fruit sampling was carried out from 2011 to 2015. Systematic random sampling was used in the two AEZs to determine the diversity of fruit flies associated with mango and guava fruits in orchards and home gardens. The mango variety “Camerounaise” and two varieties of guava (local and improved of unknown names) were available at Nkolbisson orchard. At Foumbot orchard, mango varieties included Ruby, Zill, Irwin, Julie Nyombe, Palmer, and “Camerounaise”, and as in the Nkolbisson orchard, there were local and improved guava varieties of unknown names. Twenty mature fruits each of mango and guava—based on the varieties’ maturity status—were harvested randomly from all sampling sites, and up to 10 fallen mature fruits were collected from the ground at 2-week intervals from five trees of each fruit species.

Fruits from other cultivated and wild plants were also collected during their fruiting periods from orchards, home gardens, and natural vegetation within a 70 km radius of each of the two experimental sites in WH-MR and HR-BR to determine the host range of fruit flies and the infestation levels. The number and size of fruit samples from the various plant species were primarily determined by the availability of fruits. Efforts were made to ensure a minimum collection of 20 fruits per sample at each location.

Collected fruits were classified by species, known variety, date, and sampling area, then counted and weighed. All fruits were incubated in a screenhouse (Rossel Virology Screenhouse, Clovis Lande Ass. Ltd., Kent, UK) at the IITA station in Yaoundé. Incubation units consisted of 450 mL plastic containers and 1.5 L circular plastic basins. Owing to their larger size, fruits of mango, papaya, and *Annona* spp. Were individually incubated in plastic boxes. The other fruit species were incubated in the circular plastic basins, but in groups of 3–5 depending on their size. Fruits were placed on a dome-shaped galvanized steel wire grid which rested on a 2–3 cm layer of moist heat-pasteurized Sanaga river sand as fruit flies pupariating medium. Each incubation unit was then covered with a fine-mesh cloth and secured to prevent larval escape. The incubation units were arranged on metallic shelves. The supports of each shelf were placed inside pint-size (~450 mL) containers which were maintained at full capacity with soapy water as barriers against ants and other cursorial insects. Fruit samples were incubated for up to 4 weeks to ensure that all live fruit fly larvae exited the fruits. Sand in each incubation unit was sieved twice at 12 days after the start of incubation, and at the end of incubation for the collection of fruit fly puparia. Collected puparia from each container were placed in 9 cm Petri dishes and transferred to an insectarium maintained at 25 °C, 70 ± 5% RH, and photoperiod of 12L:12D for adult emergence. Emergence dishes contained a wet mixture of table sugar and enzymatic yeast as food for full wing development of emerging adults.

All fruit fly specimens caught in traps and those that emerged from fruits were identified using fruit fly identification keys [3,16,41,42]. Voucher specimens were deposited in the IITA-Cameroon insect collection with duplicates at the IITA Biodiversity Center in Benin.

### 2.4. Statistical Analysis

Weekly counts of each species were summarized by attractant (food baits and male lures) and orchard. The total number of fruit fly specimens from traps and for each of the two genera, *Bactrocera* and *Ceratitis*, were used to calculate the relative abundance of the species. Fruit fly species diversity was estimated with the Shannon and Simpson indices using *Vegan R* 2.0. packages [26]. The Shannon index is a quantitative measure of both species richness and evenness, while the Simpson index measures evenness or species dominance [43]. Due to their non-normal distribution, the Kruskal–Wallis non-parametric test was used to compare diversity indices between orchards and attractants [28].

The seasonality data for the two experimental locations in Foumbot and Nkolbisson were summarized by each of the most abundant fruit fly species—*B. dorsalis*, *C. cosyra* and *C. anonae—*using average monthly trap catches of each species separately for each attractant (methyl eugenol, terpinyl acetate and Torula yeast).

A generalized linear model (GLM) with a Poisson error (log-link) was used to test for the effects of AEZs and years on the abundance—using weekly means—of fruit flies by species (*B. dorsalis, C. cosyra*, and *C. anonae*) and all species combined. The GLM analysis was conducted separately for Torula yeast, methyl eugenol, and terpinyl acetate. For Torula yeast, data from both sexes were pooled for the analysis.

The relationship between fruit fly catches and weather variables was explored using Pearson’s correlation [26,44]. The mean number of flies caught per month per sex over the sampling years was used in each AEZ. Species abundance data were transformed with log +1 to correct for statistical errors associated with rare or very common species [26]. Monthly temperature (minimum, mean, and maximum), relative humidity (minimum, mean, and maximum), and total monthly rainfall were calculated over the sampling years and by AEZ before analysis.

Food-bait trap catches were compared using a matched-pairs analysis [28] that identified differences in trap catches among the three food baits used in HF-BR. Similar comparisons from WH-MF were not performed since only Torula yeast was used in this AEZ. Comparisons were restricted to the three dominant species, *B. dorsalis*, *C. cosyra* and *C. anonae*, and all fruit fly species combined.

Fruit fly infestation level of each fruit sample (by fruit type and AEZ) was calculated as the number of puparia per kg of host fruits, which is a commonly used measure for estimating and comparing fruit fly infestation levels in fruits [45,46]. GLM with a Gaussian error was used to test for the effect of AEZs and sampling sites on fruit fly infestation levels by fruit and fruit fly species. Only fruit samples that were collected at least 10 times were considered in the analysis. Tukey HSD was used to compare means of fruit species infestations.

All the analyses were performed in R software version 3.6.2 [47].

## 3. Results

### 3.1. Fruit Fly Diversity and Richness in Traps

A total of 579,363 fruit fly specimens from 10 species belonging to four genera were captured in male lure and food bait traps across the two AEZs (Table 3). *Bactrocera dorsalis* was the most abundant species caught in traps (>94%), and the only species of the genus and only exotic species; followed by six *Ceratitis* species—*C. anonae*, *C. cosyra*, *C. capitata*, *C. bremii* (Guerin-Méneville), *C. ditissima* (Munro) and *C. punctata;* two *Dacus* species—*D. bivittatus* and *D. punctatifrons*; and a *Perilampsis* species. *Ceratitis anonae* was the second most abundant species, with a relative abundance of 4.10% in WH-MR, followed by *C. cosyra* which was more abundant (2.23%) in HF-BR (Table 3). The remaining species—*C. capitata*, *C. bremii*, *C. punctata*, *C. ditissima*, *D. bivittatus*, *D. punctatifrons*, and *Perilampsis* sp.—were occasionally captured in traps and comprised less than 1% of all species. Relative abundance of *B. dorsalis* and all *Ceratitis* species combined, comprised 99.0, 99.9 and 99.5% in WH-MR, HF-BR, and both AZEs combined, respectively. A large switch in the relative abundance of *C. cosyra* and *C. anonae* occurred between the two AEZs, with *C. cosyra* and *C. anonae* representing 12.6 and 85.8% of all *Ceratitis* species combined, in WH-MR, and 80.8 and 18.5%, respectively, in HF-BR.

Estimates of Shannon and Simpson diversity indices are presented in Figure 2. Fruit fly diversity was higher in WH-MR compared with HF-BR (Shannon: χ^2^_1_ = 6.55; *p* = 0.011; Simpson: χ^2^_1_ = 4.8; *p* = 0.028; Figure 2) and much more in food baits compared with male lures (Shannon: χ^2^_1_ = 8.31; *p* < 0.003; Simpson: χ^2^_1_ = 8.31; *p* < 0.003; Figure 1). The extrapolated Chao estimates for species and first-order Jackknife and Bootstrap estimates for species were higher in WH-MR (Choa: 10.6 ± 3.05; Jackknife: 10.6 ± 1.70; Bootstrap: 9.75 ± 1.01) and food baits (Chao: 9.42 ± 1.13; Jackknife: 9.83 ± 0.83; Bootstrap: 9.44 ± 0.70) compared with HF-BR (Chao: 8.41 ± 1.13; Jackknife: 8.83 ± 0.83; Bootstrap: 8.45 ± 0.66) and male lures (Chao: 8.00 ± 0.40; Jackknife: 8.83 ± 0.83; Bootstrap: 8.37 ± 0.52).

### 3.2. Abundance of the Main Fruit Flies in Traps

The overall number of *B. dorsalis* caught in methyl eugenol traps was higher (*F*_1, 564_ = 12.8; *p* < 0.001) in Nkolbisson HF-BR (488.3 ± 41.9 flies/trap) than in Foumbot WH-MR (290.4 ± 36 flies/trap), but this difference was not reflected in *B. dorsalis* counts in Torula yeast traps which were statistically similar (*F*_1, 564_ = 2.53; *p* = 0.112) in Nkolbisson HR-BR (7.44 ± 0.63 flies/trap) and Foumbot WH-MR (9.94 ± 1.41 flies/trap), respectively. Overall counts (i.e., average over the 5–6 years of trapping) of each of *C. cosyra* and *C. anonae* in terpinyl acetate differed between AEZs, but in opposite trends (*C. cosyra*: *F*_1, 569_ = 65.3; *p* < 0.001; *C. anonae*: *F*_1, 569_ = 65.6; *p* < 0.001), with *C. cosyra* counts being higher in Nkolbisson HR-BR (8.75 ± 0.84 flies/trap) than in Foumbot WH-MR (2.05 ± 0.31 flies/trap) and *C. anonae* being higher in Foumbot WH-MR (8.37 ± 1.36 flies/trap) than in Nkolbisson HR-BR (0.79 ± 0.18 flies/trap). Because of the opposite trends in trap counts of the two species, their combined counts in terpinyl acetate traps were the same (*F*_1, 569_ = 0.33; *p = 0.567*) in Foumbot WH-MR (10.6 ± 1.58 flies/trap) and Nkolbisson HR-BR (9.59 ± 0.91 flies/trap). Similarly, in Torula yeast traps, *C. cosyra* counts were significantly higher (*F*_1, 569_ = 78.2; *p* < 0.001) in Nkolbisson HR-BR (1.74 ± 0.17 flies/trap) than in Foumbot WH-MR (0.22 ± 0.06 flies/trap), while *C. anonae* and all fruit flies combined counts were significantly higher (*C. anonae*: *F*_1, 569_ = 37.2; *p* < 0.001; combined counts: *F*_1, 569_ = 14.7; *p* < 0.001) in Foumbot WH-MR (5.19 ± 1.19 flies/trap; 18.2 ± 2.49 flies/trap) than in Nkolbisson HR-BR (0.56 ± 0.08 flies/trap; 9.82 ± 0.74 flies/trap), respectively.

At Foumbot WH-MR, average yearly *B. dorsalis* counts (*F*_4, 253_ = 6.40, *p* < 0.001) in methyl eugenol and *C. cosyra* (*F*_4, 253_ = 14.0, *p* < 0.001) and *C. anonae* (*F*_4, 253_ = 12.9; *p* < 0.001) in terpinyl acetate and their combined counts (*F*_4, 253_ = 14.2, *p* < 0.001) varied substantially during the 5 years of trapping (Table 4). Similarly, *B. dorsalis*, *C. cosyra*, *C. anonae* and all trapped fruit flies combined in Torula yeast also differed among years (*B. dorsalis*: *F*_4, 253_ = 4.69; *p* < 0.001; *C. cosyra*: *F*_4, 253_ = 2.66; *p = 0.033*; *C. anonae*: *F*_4, 253_ = 13.2; *p* < 0.001; all: *F*_4, 253_ = 7.72; *p* < 0.001; Table 4). Counts of *B. dorsalis* in methyl eugenol were lowest in 2014 and 2015, and highest in 2012–2013, while in Torula yeast *B. dorsalis* was lower in 2014–2015, and higher in 2012 (Table 4). In terpinyl acetate and Torula yeast traps, *C. cosyra* counts were lowest in 2016 and 2014, and highest in 2012 and 2013, while *C. anonae* counts were lowest in 2016 and 2014–2016, and highest in 2012–2013 and 2012, respectively (Table 4).

At Nkolbisson HF-BR, average yearly *B. dorsalis* captures were similar across years in methyl eugenol (*F*_5, 307_ = 0.65; *p* = 0.661; Table 4) and in Torula yeast traps (*F*_5, 307_ = 1.35; *p* = 0.244; Table 4). In contrast, average yearly *C. cosyra* and *C. anonae* counts in terpinyl acetate traps and in Torula yeast fluctuated between years (terpinyl acetate traps: *C. cosyra*: *F*_5, 307_ = 39.2, *p* < 0.001; *C. anonae*: *F*_5, 307_ = 11.5, *p* < 0.001; Torula yeast traps: *C. cosyra*: *F*_5, 307_ = 5.90, *p* < 0.001; *C. anonae*: *F*_5, 307_ = 3.95, *p* < 0.002; Table 4). Counts of *C. cosyra* in terpinyl acetate traps were lowest in 2011 and 2013 and highest in 2015–2016, while *C. anonae* counts were lowest in 2011–2013 and highest in 2014–2016. In Torula yeast traps, *C. cosyra* counts were lowest in 2011 and 2014–2015, and highest in 2012 and 2015–2016, while *C. anonae* count trends were nearly opposite to *C. cosyra*, being highest in 2011–2013 and lowest in 2015–2016 (Table 4).

### 3.3. Comparisons of Food Baits

Comparisons of food bait catches (Table 5) revealed that Torula yeast attracted, respectively, 1.59 and 1.38-fold more *B. dorsalis* and *C. cosyra* than BioLure, while the latter attracted 2.33-fold more *C. anonae* than Torula yeast. When considering all trapped fruit flies together, Torula yeast attracted 1.28-fold more fruit flies than BioLure (Table 5). Similarly, Torula yeast trapped, respectively, 2.22 and 2.40-fold more *B. dorsalis* and *C. cosyra* than Mazoferm, while the latter and Torula yeast trapped a similar number of *C. anonae*. When considering all trapped fruit flies together, counts in Torula yeast traps were 2.13-fold more than in Mazoferm traps (Table 5).

### 3.4. Seasonal Fluctuations

The patterns of male capture across the two AZEs in successive years (2011–2016 in Nkolbisson HF-BR, and 2012–2016 in Foumbot WH-MR) using methyl eugenol and terpinyl acetate are presented in Figure 3A,B. In Foumbot WH-MR, *B. dorsalis* male populations were present throughout the year, although male numbers decreased to near zero in January and February, which coincided with the height of the dry season. In contrast, *Ceratitis* species appeared with the presence of fruits in the orchard (Figure 3A). Except in 2013, *B. dorsalis* and *C. anonae* males peaked during the rainy season in June when mango fruits were present in the orchard, but the population of both species decreased significantly in 2014 and rebounded, though not to 2012 and 2013 levels. *Ceratitis cosyra* was present in much lower densities compared with *C. anonae* in terpinyl acetate traps and *B. dorsalis* in methyl eugenol traps, but the abundance of *C. cosyra* appears to follow similar temporal patterns of *B. dorsalis* and *C. anonae*.

At Nkolbisson HF-BR, the patterns of male lure trap count also revealed the presence of *B. dorsalis* and *C. cosyra* throughout the year (Figure 3B). *Ceratitis anonae* was present in low numbers but its patterns of abundance followed that of the other two species. *Bactrocera dorsalis* exhibited peaks during the rainy season in May, which coincided with the end of mango and guava seasons (Figure 3B), and minor peaks in September during the second guava season. *Ceratitis cosyra* also followed the same trends, but the population peaks coincided with the mango season in May, and the presence of other fruit species present in the orchard, especially guava and *Annona* spp. in November–December.

The weekly inspection of the traps baited with Torula yeast allowed the evaluation of female fluctuations of *B. dorsalis*, *C. cosyra* and *C. anonae* in both AEZs as shown in Figure 4A,B. In Foumbot WH-MR, females of *B. dorsalis* and *C. anonae* were most abundant in Torula yeast traps from April through July, and peaked in June, as in male lure traps (Figure 4A). These peaks matched the presence of mango fruits in the orchard (Figure 4B). As with male lures, all fruit fly counts in Torula yeast declined in 2014–2016.

At Nkolbisson HF-BR, the period of high trap counts of *B. dorsalis* occurred from March to May and mostly peaked in April, coinciding with the full mango and guava seasons, while *C. cosyra* were more active from January to April with peaks observed in March (Figure 4B). *Ceratitis anonae* followed similar abundance patterns to *B. dorsalis*, as in Foumbot WH-MR, but occurred in much lower numbers compared with its abundance in the WH-MR AEZ.

### 3.5. Correlation of Fruit Fly Catches and Weather Variables

The catches of *C. cosyra* and *B. dorsalis* males in Nkolbisson HF-BR were positively correlated with TempMax and RHMean, respectively, while male catches of *C. cosyra* were negatively correlated with RHMin (Table 6). The rainfall data also correlated positively with the catches of all tephritids except male catches of *C. cosyra*. In Foumbot WH-MR, a positive correlation was observed between TempMin and *B. dorsalis*, TempMin and both males and females of *C. anonae*, TempMean and females of *C. cosyra*, and TempMax and females of *C. cosyra*, while a significant and negative correlation was noted between TempMean and the catches of both sexes of *B. dorsalis*, TempMax and the catches of *C. cosyra* and *C. anonae* males (Table 6). The analysis also revealed a positive and significant correlation between rainfall and *B. dorsalis* male catches (Table 6).

### 3.6. Host Range and Fruit Infestation Levels

A total of 6716 kg of fruits from 25 plant species in 16 families were collected from the two AEZs from 2011 to 2015. Mango and guava represented, respectively, 26.3 and 30.3% of the total fruits collected and incubated. Twenty-two out of 25 sampled host plant species were infested by one or more of six fruit fly species, including *B. dorsalis*, *C. cosyra*, *C. anonae*, *C. capitata*, *C. ditissima,* and *D. bivittatus* (Table 7). *Bactrocera dorsalis* was present in fruits of 21 plant species, while *D. bivittatus*, *C. capitata*, *C. ditissima*, *C. cosyra*, and *C. anonae* were present in 1, 4, 5, and 10 host plants, respectively (Table 7). Fruits of four plant species—*I. wombolu* Vermoesen (Irvingiaceae), *Dacryodes edulis* (G. Don) Lam, Burceracea), *Voacanga africana* Stapf. (Apocynaceae) and *Trichoscypha abut* Engl. (Anacardiacea)—were new host records for *B. dorsalis*.

After 5 years of sampling, five fruit fly species in the genera *Bactrocera* and *Ceratitis* were reared from mango and guava fruits across the two AEZs. *Bactrocera dorsalis*, *C. cosyra*, *C. anonae*, *C. capitata* and *C. ditissima* were the fruit fly species associated with fruit damage (Table 7). In WH-MR, *B. dorsalis* was present in 98.1 and 91% of mango and guava, respectively, while in HF-BR, *B. dorsalis* represented 99.6 and 87.2% of all fruit flies infesting mango and guava, respectively, (Table 7). The other fruit fly species had a relatively low occurrence in mango and guava in both AEZs (Table 7).

The incubation of other fruit species revealed the presence of all the fruit fly species that were collected from mango and guava, except *D. bivittatus* which principally infested papaya fruit (Table 7). Of all cultivated potential host plants, *B. dorsalis* emerged from 13 plant species, followed by *C. anonae* (7 species) in the two AEZs combined (Table 7). Of the wild potential host plants, *B. dorsalis* was also the most abundant, with occurrence in five host fruit species, out of eight sampled. *Ceratitis anonae* and *C. cosyra* were abundant on *A. muricata*, *Myrianthus arboreus* P. Beauv. and *Sarcocephalus latifolius* (Sm.) E.A.Bruce (Table 7).

Co-occurrence of *B. dorsalis* and *Ceratitis* spp. was common in all the fruit species, except in *A. carambola*, *C. limon*, *C. reticulata*, *I. gabonensis* and *I. wombolu* which were infested only by *B. dorsalis* (Table 7). Thus, *C. cosyra* co-existed with *B. dorsalis* in *S. cytherea* and *Annona montana* Macfad., while *C. anonae* and *B. dorsalis* were present in *T. abut*, banana (*Musa acuminata* Colla.), and plantain (*M.* × *paradisiaca* L.) in HF-BR (Table 7). In both AEZs, *C. cosyra*, *C. anonae,* and *B. dorsalis* were found together in mango, *A. squamosa*, *A. muricata, M. arboreus,* and *P. americana* only in HF-BR. Guava and *M. arboreus* were co-infested by *C. cosyra*, *C. anonae*, *C. capitata,* and *B. dorsalis* in HF-BR, while in both zones, *C. cosyra*, *C. anonae*, *C. capitata,* and *C. ditissima* were found with *B. dorsalis* in *P. guajava* (Table 7). *Carica papaya* was the only fruit species co-infested by *D. bivittatus*, *C. anonae* and *B. dorsalis*. This fruit was not encountered during any of the sampling periods in WH-MR.

Overall fruit infestation levels (all fruit species combined across years) were significantly (*F*_1, 1411_ = 31.5; *p* < 0.001) higher in HF-BR (48.3 ± 3.49 puparia/kg) than in WH-MR (29.4 ± 5.58 puparia/kg). In HF-BR, the number of puparia/kg of fruits also differed between the host fruit species (*F*_10, 996_ = 68.4; *p* < 0.001; Figure 5), with the highest infestation levels in *I. gabonensis*, followed by *E. japonica* and *A. muricata* with, respectively, 188.8 ± 42.1, 188.6 ± 22.4, and 187.5 ± 25.6 puparia/kg of fruits. Infestation levels of mango (39.6 ± 11.8 puparia/kg) and guava (33.4 ± 2.28 puparia/kg) in WH-MR were similar (*F*_1, 404_ = 0.62; *p* = 0.432).

## 4. Discussion

This present study was part of the fruit fly management program in mango and other orchard systems initiated in SSA following the invasion of Africa by *B. dorsalis* [6,8,13,26,48,49]. To our knowledge, it is the first and only study from the Congo basin of Central Africa to present long-term dynamics, spanning 6 years, of several species of frugivorous tephritid fruit flies by simultaneously using male lures, food baits and incubation of fruit fly host fruits. In the course of this study, we discovered four new host plant–fruit fly associations and uncovered new patterns of fruit fly population dynamics from agro-ecologies representing a cross-section of climates and conditions found in Central Africa. Our study also corroborates the findings of several other studies and provides additional information and exceptions on host fruit range, infestation rates, and seasonal dynamics of the encountered fruit fly species.

Ten fruit fly species in four genera—*Bactrocera*, *Ceratitis*, *Dacus*, and *Perilampsis*—were caught in male lure and food bait traps installed in mango and mixed-fruit orchards in the two AEZs covered by our study. Diversity analysis using Shannon and Inverse Simpson indices showed significant differences between AEZs and attractants with a higher number of species in the Nkolbisson HF-BR orchard compared with the Foumbot WH-MR orchard, probably due to higher host plant diversity in the mixed-fruit orchard than in the relatively homogeneous (mango) WH-MR orchard. Moreover, food baits, because of their broad species attraction [50] are expected to produce higher diversity indices compared with male lures—with some exceptions [27]. These comparisons are rare, however, because diversity indices are not generally calculated by attractant (e.g., male lures and food baits), which does not allow proper comparisons of diversity indices across studies. In our study, we computed diversity indices by AEZ and by male lures and the food bait Torula yeast which were used across the two AEZs. As expected, the two diversity indices—Shannon and Inverse Simpson—were larger for Torula yeast than for male lures and were considerably larger (0.5–0.9) than values reported from Western Africa by [28] for pooled data from four food baits (Torula yeast, Mazoferm, BioLure and Nulure).

Three studies across four agro-ecological zones in Western Africa [26,28,51] found 9–11 *Ceratitis* species in male lure and food bait traps placed in homogenous and mixed-fruit mango orchards over a period of 4–5 years. The differences in species occurrence and the efficiency of different lures and food baits were discussed by [28]. In our study from two Central Africa AEZs—that do not overlap with the Western Africa AEZ covered by the previous studies, we found six *Ceratitis* species in similar male lure and food bait traps. Four rare species (1–5 specimens)—*C. lentigera* Munro, *C. pedestris* (Bezzi), *C. acicularis* (Munro) and *C. penicillata* (Bigot)—from Western African mango systems were not found in our study. Moreover, three species that are generally restricted to Guinea and Sudan Savanna AEZ of Western Africa—were also absent from our study from Central Africa. Two *Dacus* species were caught occasionally in traps and probably originated from cucurbit plants in the vicinity of the trapping sites as these species are cucurbit feeders [10,26,49], but *D. bivittatus* is also known to infest *C. papaya* [1], as observed from our papaya fruit samples.

Overall, *B. dorsalis* was the dominant species, particularly in male lure traps, which is likely due to the greater attraction of methyl eugenol for this species compared with the attraction of terpinyl acetate to *Ceratitis* species [28]. The numerical dominance of *B. dorsalis,* however, was also reflected in the total number of fruit flies (>90%) that emerged from incubated mango fruits. These results are consistent with those from Eastern and Southern Africa—Kenya [16,52], Mozambique [53], Tanzania [49] and Western Africa—Benin [13,26,51], Burkina Faso [54], and Ivory Coast [55]. *Bactrocera dorsalis* was also particularly abundant in the mixed-fruit orchard at HF-BR where a large diversity of host plants was found, with several wild and cultivated fruits present all year round that could, at the same time, represent both a sink and a source for *B. dorsalis*.

Of the six *Ceratitis* species collected in our study, *C. cosyra* and *C. anonae* were the numerically dominant species in terpinyl acetate and food bait traps, but their abundance patterns were different in the two AEZs covered by our study. *Ceratitis anonae* was more abundant in the WH-MR than in HF-BR and, therefore, appears to be well adapted to the former AEZ with its generally higher altitude and cooler climate compared with HF-BR, where *C. cosyra* was more abundant than *C. anonae*. Other studies across altitude gradients in Eastern and Southern Africa have also demonstrated similar patterns of abundance for *C. cosyra* and *C. anonae* [56]. The latter also occurs in low frequency in the low altitudes of Western Africa [26,28,51]. In comparison, the abundance patterns of *C. cosyra*, which is a widespread species in Africa [57], appear to be related more to host plant availability and competition with other species [45] than to climate [58], although *C. cosyra* was shown to have a high tolerance to drier conditions based on the effects of relative humidity on survival rates of puparia (R. Hanna, unpublished data).

Food baits such as Torula yeast, Mazoferm, and under certain conditions, BioLure, are known to have broader species attraction than the specific male lures. While our study focused primarily on the diversity, seasonality, and fruit infestations of fruit flies in mango systems using primarily methyl eugenol, terpinyl acetate, Torula yeast, and fruit incubations, we used the HF-BR orchard to compare the fruit fly trapping capacity of Torula yeast, Mazoferm and BioLure. While Torula yeast is broadly more efficient in trapping fruit flies and has shown consistent efficiency in fruit fly monitoring and suppression across several AEZs [28], Mazoferm and BioLure have shown inconsistent potential in fruit fly population detection and monitoring. In South Africa, BioLure was more efficient than Torula yeast [27] probably because of the prevalence of *C. capitata* for which BioLure is an efficient attractant [32]. In our study, Torula yeast consistently caught more *B. dorsalis* and *C. cosyra* than BioLure and Mazoferm; however, BioLure was more attractive to *C. anonae* than Torula yeast which was similar to Mazoferm in attracting *C. anonae*. It is not known, however, if the same comparative trapping capacity would hold under higher *C. anonae* and *C. cosyra* abundance. While these results are consistent with those reported by [25] and [28] for the comparison of Torula yeast and BioLure, they are inconsistent with similar comparisons from other regions of Africa. In our study, Torula yeast was consistently more efficient than Mazoferm (except for *C. anonae*) which had broadly similar trapping efficiency to Torula yeast in Western and Eastern Africa. The study [59] also noted that Torula yeast was the preferred bait for detection and monitoring purposes over standard attractants such as Nulure and Mazoferm because its pH remained stable, over time, at 9.2.

The abundance of the main tephritids—*B. dorsalis*, *C. cosyra* and *C. anonae*—reported in this study varied between years and AEZs, especially in the WH-BR, where, in all the attractants, fruit fly counts in traps decreased from 2014 to 2015 to nearly 50% of the levels of the previous two years (2012–2013). The decrease in captures may be related to a combination of factors including the low production of mango varieties in the orchard due to ageing, poor tree maintenance, and the use of bait sprays in the fruit trees of the orchard surrounding the experimental block.

Long-term trapping conducted in this study established that the large within-year variations in fruit fly catches were linked to climatic conditions, but also to the direct effect of host fruit availability on the population build-up. Research paper [60] indicated that the seasonal fluctuations in most fruit fly species were characterized by high population levels during wet periods and low levels in dry periods. In our case, *B. dorsalis* and the two *Ceratitis* cycles corresponded consistently with the rainy season, between the end of April and mid-June in both AEZs. Rainfall has been reported as one of the key factors that can affect plant phenology and nutrient quality, and determine the rapid explosion of various *Bactrocera* species. Moreover, the catches of *B. dorsalis* and *Ceratitis* species males and females were found to be influenced by temperature, especially in HF-BR. Temperature is known to play an important role in the abundance of tephritids through its effect on the developmental rate, mortality, reproduction, and intensity of activity [26,29,51,61]. In HF-BR, the minimum temperature was significantly and positively correlated with male and female catches of *B. dorsalis* and *C. anonae*. This result suggested that an increase in temperature causes an increase in *B. dorsalis* and *C. anonae* populations, an observation consistent with the findings of [61] on *B. dorsalis* in South Africa. In contrast, the maximum temperature was negatively correlated with male and female catches of the three tephritids. This relationship between maximum temperature and catches of these tephritids shows the adverse impact of temperature increase on fruit fly captures and supports low populations observed in this agro-ecosystem during the dry season. The low values of the correlation coefficients obtained in the cases of significant influence of temperature, point to the involvement of other factors in this dynamic. A significant and positive correlation was obtained between rainfall and the catches of the three tephritids in the Foumbot orchard, and male catches of *B. dorsalis* in HF-BR. During this period of high humidity, between April and June, catches of these pests were more abundant, especially *B. dorsalis*. Populations of *B. dorsalis* were reported to increase with increasing rainfall [48,49]. Due to continuous rain (April-October) in the tropical forest of Central Africa, atmospheric humidity and soil moisture naturally increase, creating suitable conditions for puparia hatching and fruit development and availability that are important for fruit fly population buildup. In addition to being related to climatic conditions, the peaks of these tephritids were synchronized with the mango and guava seasons in each AEZ, especially when most of these fruits were physiologically mature. These mango and guava seasons occurred first in HF-BR and later in WH-MR. In general, host fruit availability inside orchards and the presence of alternative host fruits in the vicinity of orchards play an important role in increasing tephritid populations [12,51,62]. The availability of suitable host plants increases the number of fruits for larval development, and the alternative host plants offer survival bridges to tephritid populations in the off-seasons of the major fruit trees, and justify their presence throughout the year.

The number of puparia per kilogram of fruits varied significantly among fruit species in HF-BR, while it was the same in WH-MR. *Bactrocera dorsalis* emerged from fruits of 21 plant species belonging to 14 families, which represent about 50% of known *B. dorsalis* host plants [63]. Within this host range, the highest *B. dorsalis* infestations, in the two AEZs covered by this study, occurred in *M. indica* and *P. guajava*, two fruit species of economic importance in Cameroon and elsewhere in sub-Saharan Africa. These two major host fruits were collected in large quantities for 5 years and, therefore, provided a strong basis for confirming their suitability for *B. dorsalis*. Our findings were consistent with the results already reported by several studies from other African countries [11,13,55]. In addition to mango and guava, we recovered *B. dorsalis* from 13 other cultivated plants including *A. muricata*, *A. carambola*, *A. montana*, *A. squamosa*, *D. edulis*, *C. reticulata*, *C. limon*, *C. papaya*, *P. americana*, *M. acuminata*, *M. paradisiaca*, *E. japonica* and *S. cytherea*. These host plants constitute an important reservoir for the annual development of the pest [45]. In our study, *A. carambola*, *Citrus* spp., *C. papaya*, *P. americana* and *S. cytherea* were not favoured hosts by *B. dorsalis,* which confirms similar findings by other studies from the African continent [11,12,13]. Infestations of *Citrus* spp. by *B. dorsalis* seem to be an exception. Infestations of *Citrus sinensis* (L.) Osbeck by *B. dorsalis* in southern Benin was nearly comparable to mango infestations from the same region [64]. Differences in *Citrus* infestations by *B. dorsalis* among the various published studies could be due to varieties and other unknown factors. Similarly, in Tanzania, *A. muricata* was highly infested by *B. dorsalis* [49], whereas in this study, *A. muricata* was predominantly infested by *C. cosyra*. *Annona* species are well known as suitable hosts for several representatives of the *Ceratitis* genus [57,65]. We also identified *B. dorsalis* on *Musa* spp. as in two other studies from Africa [11,66]. Moreover, our study presents the first record of four new hosts by *B. dorsalis* including *D. edulis*, *Irvingia wombolu*, *V. africana and T. abuta*. The first three are widely distributed in Western and Central Africa, while the fourth—*T. abut*—is restricted to Central Africa [67]. Given the importance of *D. edulis* in Central Africa—owing to its nutritional value, economic role, and socio-cultural place—there is a need for a comprehensive study of *B. dorsalis* impact on the production of this important fruit species. Fruits of *I. wombulu* and the more widely distributed species *I. gabonensis* play an important role in maintaining the wild populations of the pest and its exotic parasitoid *F. arisanus* [68,69].

*Ceratitis cosyra* is a major pest of mango in sub-Saharan Africa. In our study, *C. cosyra* emerged from 10 host fruit species with the highest levels of infestation of fruits in the genus *Annona*. Infestation by *C. cosyra* was most frequent during the dry season (November–December) when *B. dorsalis* was least abundant, was notably absent in mango fruits in WH-MR, and occurred in very low numbers in fruits from HF-BR. These patterns of fruit infestations by *C. cosyra* suggest a gradual displacement of *C. cosyra* from mango fruits by *B. dorsalis* as reported in Kenya [11,70], Uganda [40], and Ghana [71] through interspecific competition [70,72]. Our studies were initiated in 2011, 7 years after *B. dorsalis* was first reported in Cameroon [13,34] and after it had become the dominant fruit fly species infesting mango. In addition to *C. cosyra*, the congeneric species *C. anonae* is known from at least 14 fruit species in equatorial Africa [65] where the fruit fly is widely distributed. In our study, *C. anonae* emerged from 13 fruit species, most of them in HF-BR, with a low number in mango and guava, and relatively high abundance in the wild fruit *M. arboreaus* and *T. abut*. Our results from guava contrast with those by [36] who reported 64% *C. anonae* emergence from guava in the HF-BR AEZ compared with ~6% obtained 7 years later in the present study. This apparent reduction in *C. anonae* populations from cultivated (guava and mango) may suggest a strong competition displacement of *C. anonae* by *B. dorsalis* which dominates fruit fly infestations of mango and guava fruits in Cameroon and elsewhere in Africa where the species occurs. In addition to host range shift, *B. dorsalis* also dominated *C. anonae* at higher altitude (WH-MR AEZ) where the latter species is well adapted. A very similar pattern was recently documented in numerous African countries where *B. dorsalis* significantly reduced host range and climatic niche of species already present, such as *C. cosyra* [11,12,70].

## 5. Conclusions

The present study of fruit fly species composition, their host range and seasonal abundance, and the use of attractants in two contrasting agro-ecological zones representing a cross-section of climates and conditions found in Central Africa, is, to our knowledge the first study to present this breadth of information from the region. Our study confirmed that host fruit availability and climate variables are the key factors that determine fluctuations in fruit fly species abundance across seasons and agro-ecological zones. The male lures methyl eugenol and terpinyl acetate were the best male attractants, respectively, of *Bactrocera* and *Ceratitis* species, and can be used, along with Torula yeast, for monitoring fruit fly abundance. Mango and guava were the major commercial host plants for *B. dorsalis*, while the non-commercial host plants *Irvingia* spp. play a key role in the persistence of *B. dorsalis* across seasons. *Bactrocera dorsalis* co-infests several host fruits with other tephritids, especially *Ceratitis* species, but they appeared in low numbers, except in *A. muricata*, *M. arboreus*, and *S. latifolius*. The information provided by this study, although limited to two agro-ecological zones, can contribute to designing an integrated fruit fly control program in Cameroon and other countries in Central Africa with similar conditions.

## Figures and Tables

**Figure 1 insects-13-01045-f001:**
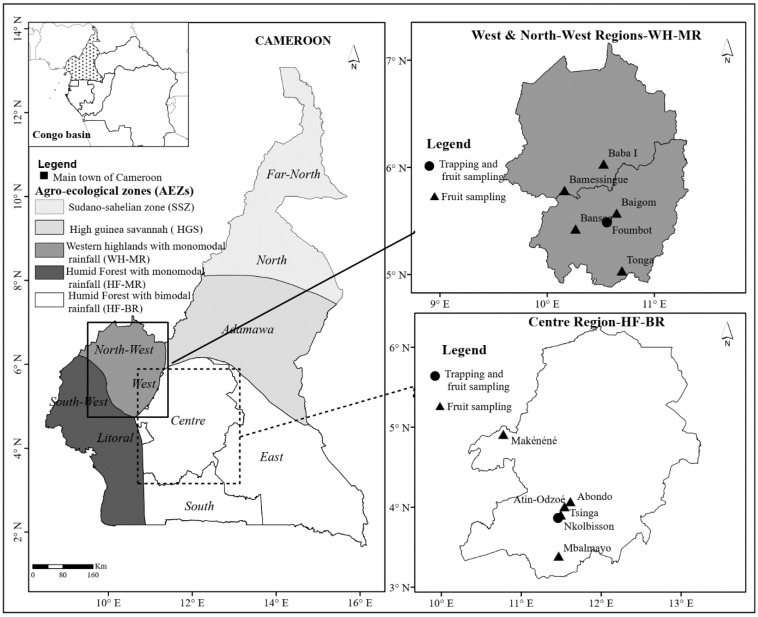
Fruit fly trapping and fruit sampling locations in Cameroon. Enlarged maps to the right correspond to the 2 agro-ecological zones—WH and HF-BR—targeted in this study.

**Figure 2 insects-13-01045-f002:**
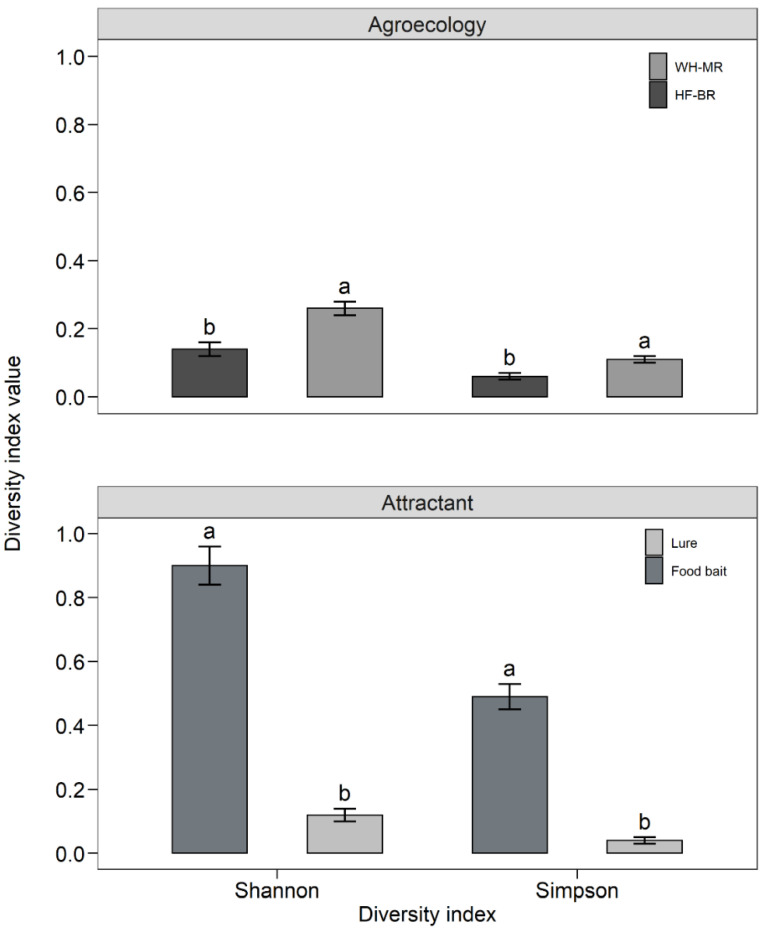
Shannon and Simpson indices (means ± SE) of fruit fly diversity according to agro-ecological zones and attractants. Means (vertical bars) followed by the same letter on each index are not significantly different (Kruskal–Wallis test, *p* < 0.05).

**Figure 3 insects-13-01045-f003:**
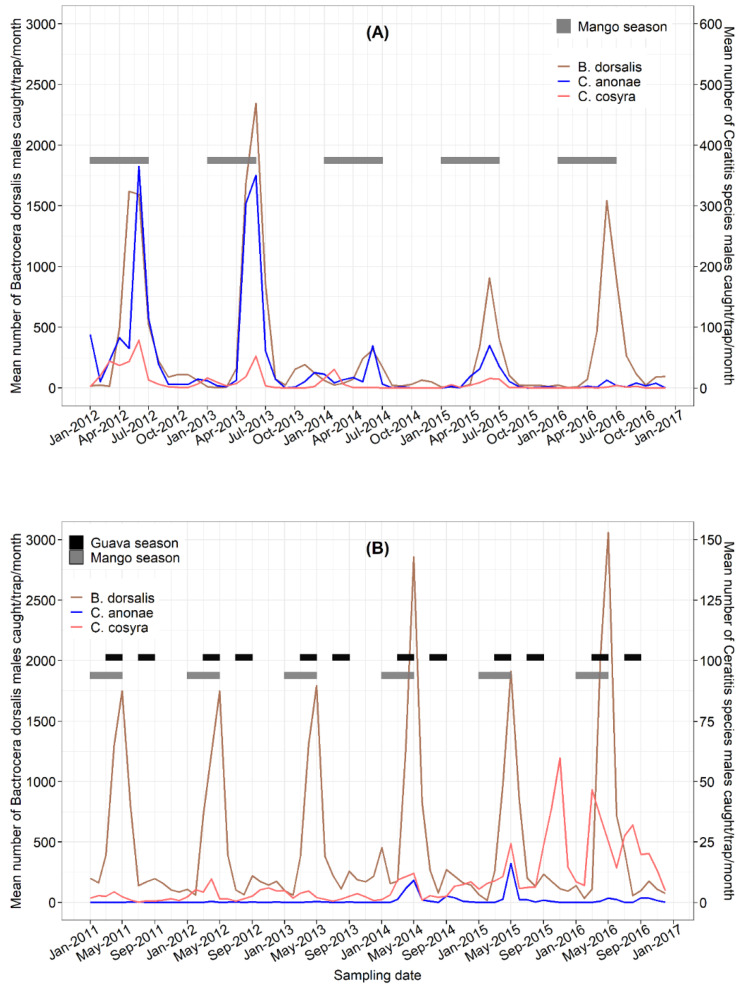
Yearly dynamics of tephritid fruit fly males in traps baited with methyl eugenol (*Bactrocera dorsalis*) and terpinyl acetate (*Ceratitis cosyra* and *C. anonae*) in Foumbot WH-MR (**A**) from January 2012 to January 2017, and Nkolbisson HF-BR (**B**) from January 2011 to December 2016, in relation to the ripening periods of mango and guava.

**Figure 4 insects-13-01045-f004:**
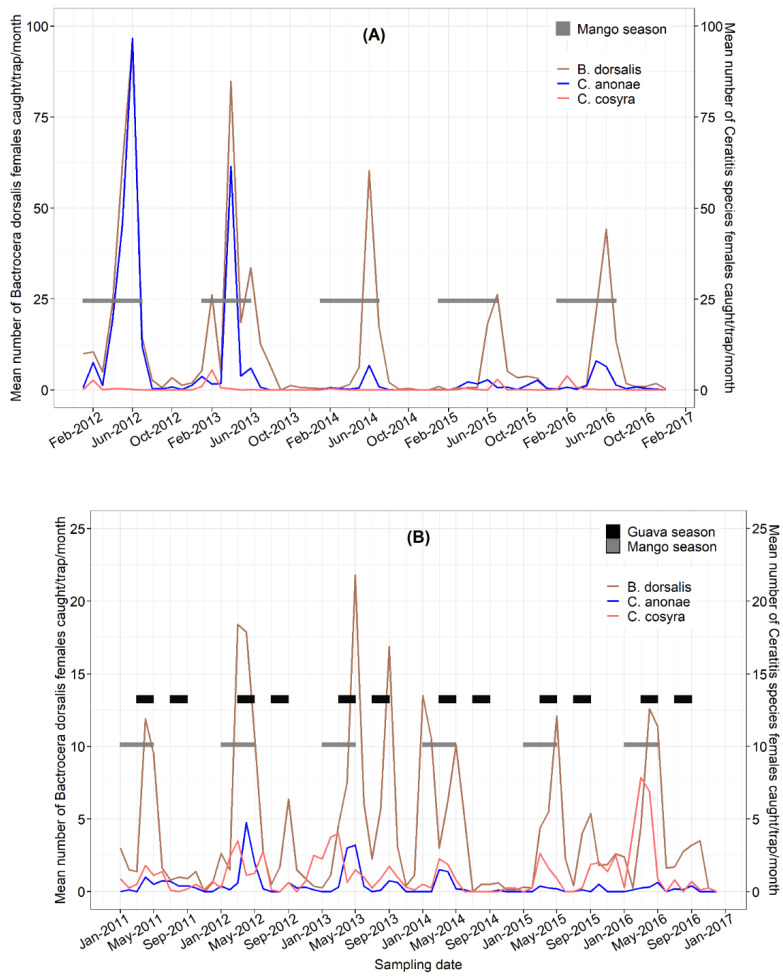
Yearly dynamics of tephritid fruit fly females in traps baited with Torula yeast in Foumbot WH-MR (**A**) from January 2012 to January 2017, and Nkolbisson HF-BR (**B**) from January 2011 to December 2016, in relation to the ripening periods of mango and guava.

**Figure 5 insects-13-01045-f005:**
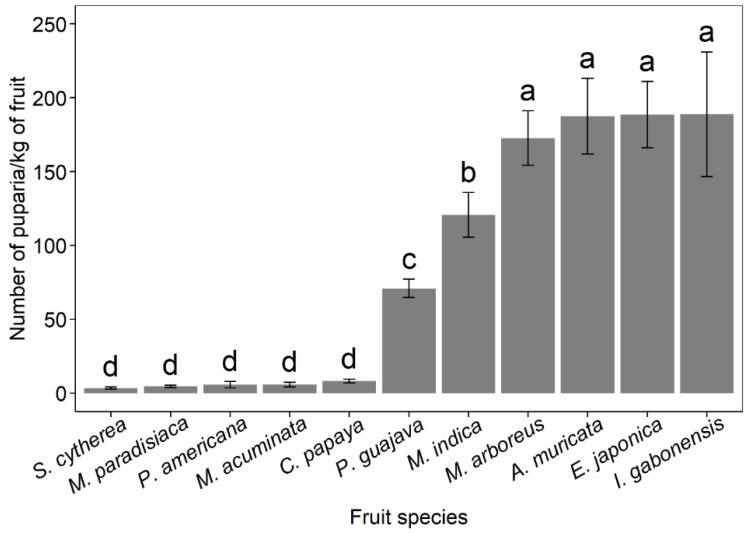
Mean infestation levels (mean number of puparia per kg of fruit ± SE) per fruit in HF-BR. Host fruit species followed by the same letter in the graph are not significantly different (GLM with Gaussian error, Tukey’s HSD, *p* < 0.05).

**Table 1 insects-13-01045-t001:** Characteristics of the experimental sites used for fruit fly trapping in two agro-ecological zones—WH and HF-BR.

Location	Description of the Area	Fruit Species	Management
Foumbot WH-MR IRAD research station	- 9 ha, of which 1 ha of mango was targeted for trapping; - A research and demonstration orchard; - Total annual rainfall of 1300 mm to 2100 mm was recorded between 2012 and 2016; - Ambient temperature of 15.3 °C to 30.5 °C was recorded between 2012 and 2016.	- Plots of mango (*M. indica*) (varieties Jolie Nyombe, Camerounaise, Ruby, Zill, Irwin, and Palmer), *Citrus* spp., guava (*P. guajava*), and avocado (Persea americana Mill (Laurales: Lauraceae)); - Various plots of vegetables with scattered single or small groups of several types of fruit trees (mostly mango and guava) in household gardens surrounded the orchard.	- Maintenance carried out by a combination of manual weeding and herbicide application.
Nkolbisson HF-BMR IITA research station	- 1 ha; - Research and demonstration orchard; - Total annual rainfall ranged from 1471 mm to 2147 mm, between 2011 and 2016; - Temperature ranged from 20.3 °C to 29.9 °C, between 2011 and 2016.	- Mango (predominantly Camerounaise variety), local and improved guava, several types of avocado, loquat, *Annona* spp. (L.) (Magnoliales: Annonaceae), hog plum (*Spondias cytherea* Sonner (Sapindales: Anacardiaceae)). - Scattered plots of cocoa, vegetables, papaya, cassava, banana, plantain, and single mango trees surrounded the experimental plot.	- Manual weeding used to control weeds in the orchard.

**Table 2 insects-13-01045-t002:** Attractants used for fruit fly trapping in two agro-ecological zones—WH and HF-BR.

Location	Trapping Period	Male Lure	Food Bait
Foumbot/WH_MR	January 2012 to December 2016.	- Methyl eugenol; - Terpinyl acetate.	Torula yeast.
Nkolbisson/HF-BMR	January 2011 to December 2016.	- Methyl eugenol; - Terpinyl acetate.	Torula yeast.
	January 2011 to December 2013.		BioLure.
	January 2012 to December 2014.		Mazoferm.

**Table 3 insects-13-01045-t003:** Abundance of fruit fly species caught in traps in two agro-ecological zones of Cameroon using three attractants during the trapping period 2011–2016.

Fruit Fly Species	WH-MR		HF-BR		Both AEZs	
	Total	Relative Abundance (%)	Total	Relative Abundance (%)	Total	Relative Abundance (%)
*Bactrocera dorsalis*	242,324	94.2	313,227	97.2	555,551	95.9
*Ceratitis cosyra*	1543	0.60	7176	2.23	8719	1.50
*Ceratitis anonae*	10,546	4.10	1641	0.51	12,187	2.10
*Ceratitis capitata*	96	0.04	49	0.02	145	0.03
*Ceratitis ditissima*	15	0.006	0	-	15	0.003
*Ceratitis punctata*	2	0.001	8	0.002	10	0.002
*Ceratitis bremii*	96	0.037	6	0.002	102	0.02
*Perilampsis* sp.	6	0.002	5	0.002	11	0.002
*Dacus bivittatus*	2555	0.99	0		2555	0.44
*Dacus punctatifrons*	0	0	68	0.02	68	0.01
Total captured	257,183	100	322,180	100	579,363	100

WH-MR: western highlands with mono-modal rainfall pattern; HF-BR: humid forest with bimodal rainfall; AEZ: agro-ecology zone.

**Table 4 insects-13-01045-t004:** Annual fruit fly species captures (means ± SE flies/trap/week) in male lures (methyl eugenol and terpinyl acetate) and food bait (Torula yeast) in 2 agro-ecological zones of Cameroon during the trapping period 2011–2016.

Trapping Year	Male Lures				Food Bait			
	Methyl Eugenol	Terpinyl Acetate			Torula Yeast			
	*B. dorsalis*	*C. cosyra*	*C. anonae*	All	*B. dorsalis*	*C. cosyra*	*C. anonae*	All
Foumbot WH-MR								
2012	412.2 ± 50.7 a	5.41 ± 0.79 a	18.9 ± 3.23 a	24.6 ± 3.81 a	19.8 ± 4.66 a (54.8)	0.28 ± 0.11 ab (54.5)	16.2 ± 4.91 a (41.9)	36.3 ± 9.34 a (49.1)
2013	464.9 ± 77.7 a	1.80 ± 0.39 b	18.9 ± 4.88 a	20.7 ± 5.16 a	15.8 ± 2.70 ab (57.9)	0.60 ± 0.22 a (52.2)	6.28 ± 2.45 b (36.8)	22.63 ± 5.73 a (52.0)
2014	88.1 ± 9.94 c	1.07 ± 0.23 bc	2.06 ± 0.42 b	3.12 ± 0.49 b	7.04 ± 2.70 c (47.0)	0.04 ± 0.03 b (55.6)	0.76 ± 0.66 c (52.2)	7.84 ± 2.92 b (47.6)
2015	161.6 ± 24.3 bc	1.11 ± 0.27 bc	5.22 ± 1.24 b	6.97 ± 1.39 b	5.29 ± 1.26 c (65.0)	0.33 ± 0.16 ab (40.4)	1.16 ± 0.22 c (36.6)	6.78 ± 1.39 b (58.7)
2016	303.3 ± 51.4 ab	0.17 ± 0.06 c	0.92 ± 0.23 b	1.09 ± 0.25 b	7.65 ± 1.96 bc (63.7)	0.51 ± 0.22 a (23.6)	1.80 ± 0.46 c (59.8)	9.97 ± 2.33 b (61.0)
Nkolbisson HF-BR								
2011	452.3 ± 57.9 a	1.59 ± 0.23 d	0.04 ± 0.02 b	1.63 ± 0.23 c	5.99 ± 0.92 a (52.3)	0.95 ± 0.16 c (33.3)	0.63 ± 0.16 ab (43.1)	7.57 ± 1.05 a (49.2)
2012	420.7 ± 54.2 a	3.72 ± 0.50 bc	0.07 ± 0.03 b	3.79 ± 0.50 c	9.35 ± 2.15 a (41.5)	2.26 ± 0.48 ab (42.1)	0.97 ± 0.28 a (23.8)	12.7 ± 2.62 a (40.2)
2013	446.0 ± 65.8 a	2.44 ± 0.36 cd	0.06 ± 0.02 b	2.50 ± 0.36 c	10.0 ± 1.43 a (39.3)	1.98 ± 0.32 ab (25.7)	0.89 ± 0.18 ab (20.4)	12.9 ± 1.62 a (39.9)
2014	598.1 ± 86.6 a	5.58 ± 0.92 b	1.96 ± 0.49 a	7.57 ± 1.31 b	7.07 ± 1.43 a (36.6)	0.76 ± 0.20 c (34.2)	0.42 ± 0.16 bcd (38.6)	8.30 ± 1.54 a (38.2)
2015	425.4 ± 60.0 a	17.75 ± 2.53 a	1.92 ± 0.61 a	19.8 ± 2.74 a	5.88 ± 0.92 a (42.1)	1.39 ± 0.27 bc (19.7)	0.19 ± 0.06 d (35.0)	7.62 ± 1.04 a (37.7)
2016	588.4 ± 109.8 a	21.27 ± 2.36 a	0.64 ± 0.16 b	22.0 ± 2.43 a	6.39 ± 1.24 a (41.7)	3.13 ± 0.76 a (41.1)	0.24 ± 0.08 cd (28.0)	9.85 ± 1.67 a (41.1)

WH-MR: western highlands with a mono-modal rainfall pattern; HF-BR: humid forest with bimodal rainfall. Values in parenthesis are percentages for males. Mean values (±SE) in the same column followed by the same letter are not significantly different in each agro-ecological zone (GLM with quasi-Poisson distribution, Tukey’s HSD, 0.05).

**Table 5 insects-13-01045-t005:** Comparison of fruit fly responses (means ± SE flies/trap/week) to BioLure vs. Torula yeast, and Mazoferm vs. Torula yeast, in the Nkolbisson orchard (humid forest with bimodal rainfall).

Attractant	*Bactrocera dorsalis*	*Ceratitis cosyra*	*Ceratitis anonae*	All Fruit Flies
BioLure	5.33 ± 0.52	1.25 ± 0.16	1.94 ± 0.34	8.66 ± 0.81
Torula yeast	8.45 ± 0.92	1.73 ± 0.20	0.83 ± 0.12	11.1 ± 1.09
t-ratio, df, *p*-value	3.36, 155, <0.001	2.18, 155, 0.030	−3.48, 155, <0.001	2.46, 155, 0.016
Mazoferm-6%	3.96 ± 0.56	0.69 ± 0.10	0.63 ± 0.11	5.30 ± 0.68
Torula yeast	8.81 ± 0.98	1.67 ± 0.21	0.76 ± 0.12	11.3 ± 1.15
t-ratio, df, *p*-value	4.40, 155, <0.001	4.60, 155, <0.001	1.30, 155, 0.194	4.75, 155, <0.001

All the differences were significant (*p* < 0.05), according to the indicated probability values corresponding with the t-ratio and degrees of freedom following the matched pairs analysis.

**Table 6 insects-13-01045-t006:** Spearman’s correlation coefficient between weather variables and fruit fly species commonly caught per trap per month in the two agro-ecological zones.

Weather Variables	*Bactrocera dorsalis*		*Ceratitis cosyra*		*Ceratitis anonae*	
	Male	Female	Male	Female	Male	Female
Foumbot WH-MR						
TempMin	0.422 **(0.001)**	0.450 **(0.001)**	0.186 (0.154)	0.025 (0.848)	0.336 **(0.009)**	0.372 **(0.003)**
TempMean	−0.331 **(0.010)**	0.033 (0.804)	−0.062 (0.636)	0.442 **(0.001)**	−0.256 (0.049)	0.117 (0.375)
TempMax	−0.496 **(0.001)**	−0.270 **(0.037)**	−0.300 **(0.020)**	0.294 **(0.023)**	−0.510 **(0.001)**	−0.203 (0.119)
RHMin	0.122 (0.352)	0.119 (0.365)	0.070 (0.596)	−0.025 (0.852)	0.050 (0.703)	0.007 (0.955)
RHMean	0.129 (0.324)	0.069 (0.600)	0.028 (0.830)	−0.118 (0.371)	0.032 (0.810)	−0.059 (0.652)
RHMax	0.088 (0.505)	0.019 (0.887)	0.076 (0.565)	−0.120 (0.362)	0.134 (0.309)	0.001 (0.998)
Rainfall	0.298 **(0.021)**	0.132 (0.313)	−0.187 (0.154)	−0.199 (0.127)	−0.042 (0.750)	−0.042 (0.750)
Nkolbisson HF-BR						
TempMin	0.145 (0.223)	0.079 (0.507)	−0.179 (0.131)	−0.079 (0.509)	−0.079 (0.509)	−0.079 (0.509)
TempMean	−0.003 (0.980)	0.190 (0.109)	0.171 (0.152)	−0.037 (0.756)	−0.037 (0.756)	−0.037 (0.756)
TempMax	−0.095 (0.429)	0.063 (0.598)	0.397 **(0.001)**	0.146 (0.221)	0.146 (0.221)	0.146 (0.221)
RHMin	0.268 (0.023)	0.054 (0.651)	−0.247 **(0.036)**	0.111 (0.354)	0.111 (0.354)	0.111 (0.354)
RHMean	0.267 **(0.024)**	0.062 (0.607)	−0.156 (0.191)	0.178 (0.134)	0.178 (0.134)	0.178 (0.134)
RHMax	0.145 (0.225)	0.077 (0.523)	−0.085 (0.478)	0.137 (0.252)	0.137 (0.252)	0.137 (0.252)
Rainfall	0.535 **(0.001)**	0.374 **(0.001)**	0.209 (0.078)	0.429 **(0.001)**	0.429 **(0.001)**	0.429 **(0.001)**

WH-MR: western highlands with a mono-modal rainfall pattern; HF-BR: humid forest with bimodal rainfall. Values in parenthesis are *p*-values (0.05), with significant values in bold.

**Table 7 insects-13-01045-t007:** Host fruits species sampled from 2011 to 2015 and abundance of fruit fly species that emerged across two agro-ecological zones of Cameroon.

Plant Family/Species	Origin	Sampling Site	Weight (kg)	No. of Emerged Adult Fruit Flies *					
				*B. dorsalis*	*C. cosyra*	*C. anonae*	*C. capitata*	*C. ditissima*	*D. bivittatus*
Western Highlands with Monomodal Rainfall									
Anacardiaceae									
*Mangifera indica* L.	Exotic	Baba I, Bamessingue, Bansoa, Foumbot	964	14,462 (98.1)	7 (0.05)	269 (1.83)	-	-	-
Lauraceae					-	-	-	-	-
*Persea americana* Miller	Exotic	Foumbot	71.4	14 (100)	-	-	-	-	-
Myrtaceae									
*Psidium guajava* L.	Exotic	Baba I, Bamessingue, Bansoa, Foumbot	1186	22,058 (91.0)	30 (0.12)	2124 (8.76)	9 (0.04)	17 (0.07)	-
Rosaceae									
*Eriobotrya japonica* (Thunb.) Lindley	Exotic	Santa	3.0	415 (99.5)	-	-	-	2 (0.48)	-
Rubiaceae									
*Sarcocephalus latifolius* (Smith) Bruce	Native	Tonga	6.03	-	314 (100)	-	-	-	-
Rutaceae									
*Citrus limon* (L.) Baufman f.	Exotic	Foumbot	3.10	-	-	-	-	-	-
*Citrus reticulata* Blanco	Exotic	Foumbot	3.20	17 (100)	-	-	-	-	-
*Citrus sinensis* (L.) Osbeck	Exotic	Foumbot	5.35	-	-	-	-	-	-
**Plant Family/Species**	**Origin**	**Sampling Site**	**Weight (kg)**	**No. of Emerged Adult Fruit Flies ***					
				** *B. dorsalis* **	** *C. cosyra* **	** *C. anonae* **	** *C. capitata* **	** *C. ditissima* **	** *D. bivittatus* **
**Humid Forest with Bimodal Rainfall**									
Anacardiaceae									
*Mangifera indica* L.	Exotic	Nkolbisson	800	43,075 (99.6)	26 (0.06)	140 (0.32)	4 (0.01)	-	-
*Spondias cytherea* Sonner	Exotic	Nkolbisson	108	126 (99.2)	1 (0.79)	-	-	-	-
** *Trichoscypha abut* ** **Engl.**	Native	Atin-Odzoé	0.39	1 (14.3)	-	6 (85.7)	-	-	-
Annonaceae									
*Annona montana* L.	Exotic	Tsinga	6.44	3 (11.5)	23 (88.5)	-	-	-	-
*Annona muricata* L.	Exotic	Nkolbisson	96	1159 (9.22)	11,265 (89.7)	142 (1.13)	-	-	-
*Annona squamosa* L.	Exotic	Nkolbisson	164	2406 (55.6)	961 (22.2)	960 (22.2)	-	-	-
Apocynaceae									
** *Voacanga africana* ** **Stapf.**	Native	Tsinga, Nkolbisson	1.1	1 (14.3)	-	-	-	6 (85.7)	-
Burseraceae									
** *Dacryodes edulis* ** **(G. Don) Lam**	Native	Nkolbisson	27	17 (100)	-	-	-	-	-
Caricaceae									
*Carica papaya* L.	Exotic	Nkolbisson	628	2436 (77.8)	-	8 (0.26)	-	-	687 (21.9)
Cercopriaceae									
*Myrianthus arboreus-*P.-Beauv	Native	Nkolbisson	142	99 (1)	1 (0.01)	9803 (98.8)	-	15 (0.15)	-
Clusiaceae									
*Allanblackia floribunda* Oliv.	Exotic	Mbalmayo	5.3						
*Garcinia xanthochymus* Hook. f.	Exotic	Mbalmayo	0.06	-	-	-	10 (100)	-	-
Irvingiaceae									
*Irvingia gabonensis* (Aubry Lecomte) Baill	Native	Mbalmayo, Nkolbisson	138	20,775 (100)	-	-	-	-	-
** *Irvingia wombolu* ** **Vermoesen**	Native	Abondo	79	6391 (100)	-	-	-	-	-
**Plant Family/Species**	**Origin**	**Sampling Site**	**Weight (kg)**	**No. of Emerged Adult Fruit Flies ***					
				** *B. dorsalis* **	** *C. cosyra* **	** *C. anonae* **	** *C. capitata* **	** *C. ditissima* **	** *D. bivittatus* **
**Humid Forest with Bimodal Rainfall**									
Lauraceae									
*Persea americana* Miller	Exotic	Nkolbisson	287	227 (38)	353 (59.0)	18 (3.01)	-	-	-
Moraceae									
*Ficus mucuso* Filcalho	Native	Nkolbisson	20.5	-	-	-	-	-	-
Musaceae									
*Musa acuminate* L.	Exotic	Nkolbisson	479.5	1421 (99.7)	-	4 (0.28)	-	-	-
*Musa paradisiaca* L.	Exotic	Nkolbisson	564.3	1778 (99.2)	-	15 (0.84)	-	-	-
Myrtaceae									
*Psidium guajava* L.	Exotic	Nkolbisson	849.6	34,619 (87.2)	2392 (6.03)	2569 (6.47)	98 (0.25)	7 (0.02)	-
Oxalidaceae									
*Averrhoa carambola* L.	Exotic	Tsinga	12.7	3 (100)	-	-	-	-	-
Rosaceae									
*Eriobotrya japonica* (Thunb.) Lindley	Exotic	Nkolbisson	32.4	1413 (56.3)	5 (0.2)	1070 (42.6)	24 (0.96)	-	-
Rubiaceae									
*Sarcocephalus latifolius* (Smith) Bruce	Native	Makénéné	9.69	281 (29.2)	681 (70.8)	-	-	-	-
Rutaceae									
*Citrus limon* (L.) Baufman f.	Exotic	Nkolbisson	20.9	7 (100)	-	-	-	-	-
*Citrus reticulata* Blanco	Exotic	Nkolbisson	1.43	-	-	-	-	-	-

* Values in parenthesis are percentages of the adults of a species emerging from all the incubated fruits of a fruit species. Host fruits species in bold, are new records for *B. dorsalis*.

## Data Availability

The data presented in this study are available at https://doi.org/10.25502/2fr3-pd51/d and https://doi.org/10.25502/513v-yg51/d.

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
