# Peer review of "Tephritid Fruit Fly Species Composition, Seasonality, and Fruit Infestations in Two Central African Agro-Ecological Zones"

_insects, 2022, doi:10.3390/insects13111045_

Round 1

Reviewer 1 Report (Previous Reviewer 1)

In my case, the authors complied with all changes requested. I think the authors increased the quality of the manuscript with all the adjustments they made, including the addition of two tables to describe the methodology.

I highlighted a few minor mistakes I discovered in the manuscript. I attach the PDF.

Author Response

Comments and Suggestions for Authors

In my case, the authors complied with all changes requested. I think the authors increased the quality of the manuscript with all the adjustments they made, including the addition of two tables to describe the methodology.

I highlighted a few minor mistakes I discovered in the manuscript.

Response: minor mistakes corrected.

Reviewer 2 Report (Previous Reviewer 2)

The revised manuscript has addressed most of the issues i had in the original review amnd is now acceptable for publication

Author Response

Comments from Reviewer No. 2

The revised manuscript has addressed most of the issues i had in the original review and is now acceptable for publication

Response: Thank you for your revision.

This manuscript is a resubmission of an earlier submission. The following is a list of the peer review reports and author responses from that submission.

Round 1

Reviewer 1 Report

Nanga Nanga et al. present a very comprehensive study on the population dynamics of various species of fruit flies, including some pests considered to be economically important, conducted over a period of 5-6 years in two agro-ecological zones (AEZs), Western Highlands with a mono-modal rainfall (WH-MR) and Humid Forest with bimodal-rainfall (HF-BR) in Central Africa (Cameroon). The authors present systematic data of fruit flies catches with different attractants and baits, as well as data on infestation in various fruits. The data on infestation and captured flies were analyzed by AEZ based on the type of attractant or bait used, the fruit seasonality (emphasis in mango and guava), climatological factors (temperatures, rainfall, etc.), and diversity indices by region over the course of 5-6 years. The results are well presented and understandable.  Discussion and conclusion are well.  

I have just next minor revisions:

L162- Specify if mango ochard had management.

L287,446,450,451,454,455,458. The abbreviation for HF-BR is incorrect along those lines.

L493- Add the female:male proportion in Table 2 below the means. Food baits attract both females and males, so this information will be useful in future studies for comparison.

L560 - Add the AZE abbreviation to the orchard name in figures 3 and 4 (for example, Foumbot – WH-MR). The authors have used WH-MR or HF-BR throughout the manuscript, and readers are unfamiliar with the orchard name until this point.

References section needs depuration (e.g. Bactrocera Invadens).

Reviewer 2 Report

This paper reports on a study on the presence, distribution and

host status of tephritid fruit fly species in central Africa (Cameroon)

and is part of a PhD study by the senior author. It is generally well written and the 

results and conclusions are in line with the intent of the study

The research follows well established protocols and is not particularly novel

and the majority of the results in line with options of experts in the field

The inclusion of male lures in this study does not add much to the study give

that these lures are known to be highly attractive to some of the species

 Reporting the male capture from the food baits would be more useful in discussion

of species abundance and diversity. The variation in population over several years is particularly noteworthy

the amount of overall data collected is impressive as is the discussion on diversity indices

The authors should describe how they randomly sampled the fruit collection as frequently these types of stdies

are biased towards infested fruits and the authors do not include non infested frui data

Also it would have been good to sample immature and mature-green fruit as opposed to only ripe fruits

The authors should reference the known issue of B. dorsalis competition with other species due toboth elevation, tempand rainfall

competitive displacement will also bias the abundance of particular species in a given host fruit

Because this is one of the few studies of this type on the region and the findingof

a few new host records the study is worthy of publication. With minor revisions 

Reviewer 3 Report

Major concerns

The manuscript is too redundant and needs further refinement.

Minor concerns

1、The “introduction” needs to be improved, as some contents are redundancy while some are lacked. For example, paragraph 1-4 can be compressed into one part to describe the current situation of Tephritid fruit fly in Africa.

2、Part 2.1: The description of this paragraph is confusing. A better way is to list the site information in a table.

3、Part 2.2 The description of this paragraph is confusing. Please briefly describe your trap assay with a figure.

4、“Statistical analysis” a better way is to show the statistical analysis methods in your figures and tables. For this part, you just need to briefly mention the method you used for analysis.

5、 The figures in your article occupy too much space.